# Application of various mixtures of medicinal herbs in the diet of laying hens: Evaluating preventive approach of fatty liver syndrome

Mohammadreza Khodaei[1], Mehran Torki[1]*, Fariborz Khajali[2], Iraj Karimi[3]

1 Department of Animal Science, College of Agriculture and Natural Resources, Razi University, Kermanshah, Iran, 2 Department of Animal Science, Shahrekord University, Shahrekord, Iran, 3 Department of Pathology, Faculty of Veterinary Medicine, Shahrekord University, Shahrekord, Iran

* torki@razi.ac.ir

## Abstract

Fatty liver hemorrhagic syndrome (FLHS) is a major health issue in laying hens, which is associated with reduced productive performance and increased mortality. This study investigated the efficacy of three herbal additive mixes as dietary interventions to prevent FLHS in a total of 384 LSL-Lohmann laying hens from 65 to 77 weeks of age. Hens were allocated to eight treatments in a 2 × 4 factorial design, comprising two basal diets—a standard diet (SD) and a high-energy, low-protein challenge diet (CD)—alongside four dietary interventions (three herbal mixes and a control). The herbal mixes were formulated as follows: Mix 1 (turmeric, fumitory, green tea, milk thistle), Mix 2 (lemon, black pepper, sumac, chicory), and Mix 3 (garlic, artichoke, ginger, shallot). Productive performance, blood variables, and liver lesion scores were carefully assessed. The results demonstrated that the CD decreased feed intake (FI; P = 0.0001), egg production (EP; P = 0.0001), and egg weight (EW; P = 0.0001) from week 5 onward. Birds received the CD had poorer feed conversion ratio (FCR) as opposed to the SD in weeks 5 and 6 of the trial. Feeding the CD resulted in higher circulatory levels of aspartate aminotransferase (AST; P = 0.0001) and triglycerides (TG; P = 0.0001), and a higher frequency of severe livers scores (P = 0.002). Notably, Mix 2 emerged as the most effective intervention, significantly reversing the negative impacts of the CD across all measured parameters. However, the herbal mixes did not significantly affect heterophil and lymphocyte counts or hemoglobin (Hb) concentration when fed with the CD. Overall, the CD significantly impaired productive performance and increased liver pathological lesions. However, supplementing the CD with Mix 2 (lemon, black pepper, sumac, and chicory) consistently led to the most significant improvements across all measured parameters, proving to be an effective dietary intervention to prevent FLHS in laying hens.

**Data availability statement:** All relevant data are within the manuscript and its Supporting Information files.

**Funding:** The author(s) received no specific funding for this work.

**Competing interests:** The authors have declared that no competing interests exist.

## Introduction

A typical layer diet provides approximately 2–4 g of lipids per day, which is considerably less than the amount required for body weight gain and egg formation [1]. In birds, lipids are primarily synthesized in the liver. As a result, there is a heavy burden on the liver to form necessary lipids. Consequently, dealing with an enormous amount of lipids predisposes laying hens to fatty liver hemorrhagic syndrome (FLHS) [2,3].

FLHS is a non-infectious disease characterized by the excessive accumulation of fat in the liver and abdominal cavity. This significant fat buildup can lead to hepatocyte rupture, internal bleeding, and sudden death in affected birds [4,5]. FLHS is more commonly observed in laying hens that are housed in cage systems [5,6]. Affected flocks suffer substantial economic losses due to decreased egg production (EP) and increased mortality rates [4,7]. During necropsy, birds that died from FLHS exhibit enlarged, soft, yellow, and fatty livers. This condition makes the liver more susceptible to tearing and bleeding [3,5]. The exact cause of FLHS remains unclear, but several factors are thought to contribute. These include nutritional factors like excess dietary energy, fungal toxins, and anti-nutritional substances [2,8,9], as well as genetic predisposition, and environmental factors such as ambient temperature and rearing system [5,10]. It is likely that a combination of these factors may contribute to the development of FLHS.

Given the multifactorial nature of FLHS, various nutritional strategies have been explored to mitigate its impact—particularly those aimed at reducing hepatic lipid synthesis [6]. Among these, medicinal plants have emerged as a promising approach due to their bioactive compounds with health-promoting properties [11–21]. These compounds often include flavonoids, alkaloids, sulfur compounds, and dietary fibers, all of which possess documented antioxidant, anti-inflammatory, and lipid-lowering properties [22]. Specific examples of these plants and their key bioactive constituents are summarized in Table 1.

Beyond their physiological benefits, herbal interventions also contribute to improved animal welfare by alleviating the clinical manifestations of FLHS and reducing mortality [4,7]. As global egg demand continues to rise, there is increasing pressure on poultry producers to enhance the health, productivity, and welfare of laying hens in a sustainable manner. However, the prevalence of metabolic disorders such as FLHS poses a major challenge to achieving these goals, especially in intensive production systems. Given the limitations of current management and nutritional strategies, there is a critical need to identify effective, natural interventions that can mitigate FLHS without compromising productivity.

While individual medicinal herbs have been examined for their potential benefits in laying hens, the efficacy of combinations of these herbal additives in preventing or mitigating FLHS has not been thoroughly investigated. Thus, the objectives of the present study were to develop an appropriate formulation of herbal additives in laying hen diets to evaluate productive responses, blood parameters, and the development of FLHS.

**Table 1. Medicinal plants and their key bioactive compounds.**

| Plant | Scientific name | Main bioactive compounds | References |
|---|---|---|---|
| Sumac | *Rhus coriaria* | Flavonoids, anthocyanins | [11] |
| Lemon | *Citrus limon* | Vitamin C, flavonoids, limonoids | [12] |
| Black pepper | *Piper nigrum* | Piperine alkaloids | [13] |
| Chicory | *Cichorium intybus L.* | Inulin, sesquiterpenes | [14] |
| Milk thistle | *Silybum marianum* | Silymarin (flavonoid complex) | [15] |
| Garlic/ Shallot | *Allium sativum, A. cepa* | Allicin (organosulfur compound) | [16] |
| Artichoke | *Cynara scolymus* | Inulin, oligofructose | [17] |
| Green tea | *Camellia sinensis* | Catechins (flavonoids) | [18] |
| Turmeric | *Curcuma longa* | Curcumin | [19] |
| Fumitory | *Fumaria officinalis* | Protopine, fumaric acid | [20] |
| Ginger | *Zingiber officinale* | Gingerol, gingerdiol, gingerdione | [21] |

## Materials and methods

### Ethics approval

The Animal Ethics Committee of Razi University, Kermanshah, Iran approved guidelines were followed for all experimental protocols (IR.RAZI.REC.1400.051). These guidelines adhered to the EU standards for the protection of animals and/or feed legislation.

### Experimental design and dietary treatments

A number of 1000 fertile eggs (60.5±3.5 g, Lohmann LSL-Lite strain) were obtained from a local breeding company (Toyoor-Barekat, Tehran, Iran). The eggs were collected from a breeder flock at 45 weeks of age. Fertile eggs were set in the laboratory incubator (MG-1000 model, Rcom Co., Ltd., Korea) of the Animal Science Department, Razi University, Kermanshah, Iran. Hatched chicks were then raised according to the strain guideline until the onset of the experimental trial at the age of 65 weeks [23]. A total of 384 laying hens were randomly selected from the flock. In doing so, all hens were weighed to obtain an average body weight in the flock (1,605±85 g). The hens were housed in wire cages (Machine Toyour, Co., Juybar, Mazandaran, Iran) measuring 45×45×45 cm, with three birds per cage, corresponding to a stocking density of 675 cm² per bird. The experiment followed a 2×4 factorial arrangement in a completely randomized design (CRD), with treatments randomly assigned to eight replicates (each consisting of two adjacent cages containing six birds in total). Sample size was determined based on expected effect sizes from previous studies to achieve 80% power at α=0.05. To minimize potential bias, all data collection and subsequent laboratory analyses were performed by personnel blinded to the dietary treatments. A 16-hour light/8-hour dark lighting schedule was implemented using white LED lighting (Namanoor Co., Tehran, Iran), providing an intensity of approximately 10–15 lux at bird level. The temperature-humidity index (THI) over the 12-week period is presented in Fig 1. Hens had free access to feed and water throughout the experiment.

Two basal diets were formulated for the experiment: a standard diet (SD) formulated according to the strain's recommended guidelines (2500 kcal/kg AME and 15% crude protein) [23], and a challenge diet (CD) having high energy and low protein contents (3100 kcal/kg AME and 12% crude protein) designed to induce FLHS. Within each basal diet, there were four dietary interventions including three herbal mixes along with a control with no herbal additives. Mix 1 included turmeric, fumitory, green tea, and milk thistle. Mix 2 contained lemon, black pepper, sumac, and chicory. Mix 3 comprised garlic, artichoke, ginger, and shallot. The experimental diets and their chemical composition are presented in Table 2. A two-week adaptation period was implemented to acclimate hens to the diets prior to the 12-week experimental trial. The

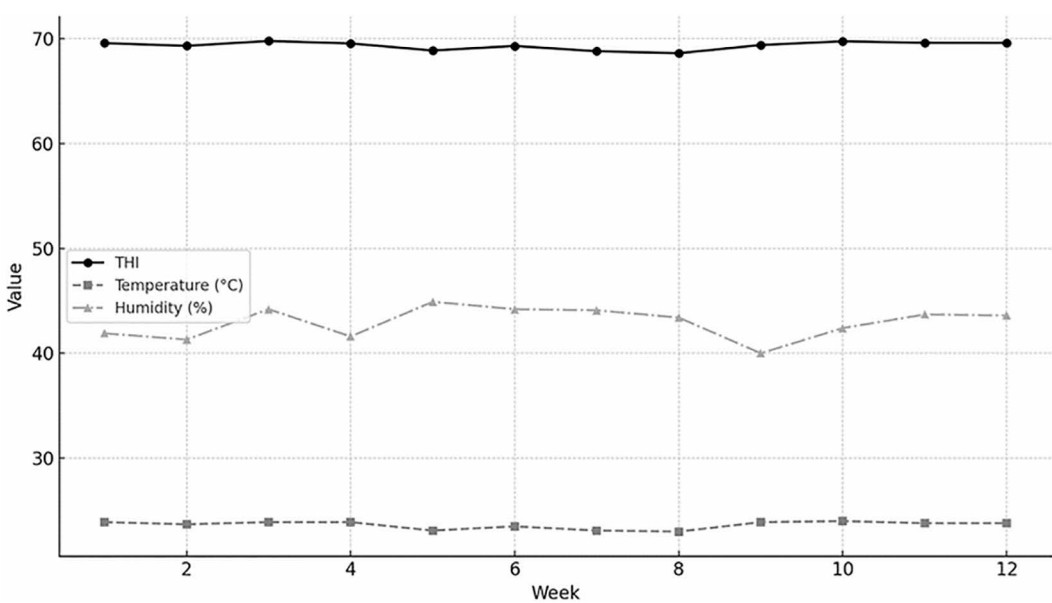

**Fig 1. Weekly variation in temperature-humidity index (THI), ambient temperature, and relative humidity during a 12-week experimental period.** THI was calculated using the formula: THI = 0.8 × $T_{db}$ + RH × ($T_{db}$ − 14.3)/100 + 46.3 [24], where $T_{db}$ is the dry-bulb temperature (°C) and RH is the relative humidity (%).

adaptation period was considered to minimize the error effect caused by the changes in the diets. The experimental trial lasted for 12 weeks (weeks 65–77).

## Measurements

**Gas chromatography-mass spectrometry (GC-MS).** Volatile compounds of herbal samples were identified using a headspace (HS) sampling technique followed by GC-MS analysis. A qualitative GC-MS analysis was performed with an HP 6890 gas chromatograph interfaced with an HP 5975C Mass Selective Detector (Agilent Technologies, Foster City, CA, USA). The analysis employed electron ionization (EI) mode at 70 eV and followed the procedure described by Naghneh et al. [25]. The composition of the medicinal herbs used as dietary feed additives has shown in additional file S1 File. The excel file named as "Khodaei.xlsx" S2 File, is the pure data of the field animal productive performance data before statistical analysis.

**Productive performance.** Egg count and weight were recorded on a daily basis, and these data were used to calculate hen-day egg production (EP), and egg weight (EW). Moreover, feed conversion ratio (FCR) was calculated based on feed intake (FI) and egg performance data [26].

**Blood parameters.** At weeks 65 (the start), 70 (the middle) and 77 (the end of trial), two laying hens from each replicate (16 hens per treatment) were randomly selected for blood collection from the wing vein. A sample of 3 ml blood was collected in a test tube without any anticoagulant. Blood samples were immediately centrifuged at 3000 rpm for 10 minutes and the resulting sera were stored at −20°C until laboratory tests. The following parameters were determined according to the manufacturer's instructions using commercial diagnostic kits (Pars Azmun, Iran): aspartate aminotransferase (AST), total protein (TP), total triglyceride (TG), total cholesterol (TC), low-density cholesterol (LDL), high-density cholesterol (HDL), uric acid (UA), and hemoglobin (Hb) (Parsazmun, Tehran, Iran) [27]. In addition, a smear of blood was prepared from each blood sample to count heterophiles and lymphocytes.

**Table 2. Ingredients and composition of the treatment diets.**

| Ingredients (%) | SD | CD |
|---|---|---|
| Corn | 57 | 66 |
| Soybean meal | 24 | 17 |
| Wheat bran | 6.4 | – |
| Soybean oil | 1 | 6 |
| Methionine | 0.15 | 0.15 |
| Salt | 0.2 | 0.2 |
| Sodium bicarbonate | 0.1 | 0.1 |
| Vitamin premix[a] | 0.25 | 0.25 |
| Mineral premix[b] | 0.25 | 0.25 |
| Oyster shell | 9.2 | 9.2 |
| Dicalcium phosphate | 1.5 | 1.5 |
| Multi enzyme containing phytase | 0.05 | 0.05 |
| **Nutrient composition** | | |
| ME (Kcal/kg) | 2500 | 3100 |
| Crude protein (%) | 15 | 12 |
| Crude fiber (%) | 2.1 | 2.1 |
| Calcium (%) | 3.7 | 3.7 |
| Sodium (%) | 0.13 | 0.13 |
| Available phosphorus (%) | 0.36 | 0.36 |

Treatment diets: SD (standard diet = normal protein and normal ME); CD (challenge diet = low protein and high ME).

[a]Vitamin mixture per 0.3 kg/100 kg of diet: Vitamin A, 7,700,000 IU; Vitamin D3, 3,300,000 IU; Vitamin E, 6600 mg; VitaminK3, 550 mg; thiamine,2200 mg; riboflavin, 4400 mg; Vitamin B6, 4400 mg; capantothenate, 550 mg; nicotinic acid, 200 mg; folic acid, 110 mg; choline chloride, 275,000 mg; biotin,55 mg; Vitamin B12, 8.8 mg.

[b]Mineral mixture per 0.3 kg/100 kg of diet: Mn, 66,000 mg; Zn, 66,000 mg; Fe, 33,000 mg; Cu, 8,800 mg; Se, 300 mg.

**Hepatic histopathology and scoring system.** During the experimental period, all mortalities were weighed and necropsied, and the incidence of FLHS was assessed in all treatment groups. At the end of trial (Week 77), one hen from each cage (8 hens per treatment) were randomly selected, anaesthetized via inhalation of 60% $CO_2$, and then sacrificed by cervical dislocation. $CO_2$ was delivered from compressed gas canister and delivered using a gradual fill method with a displacement rate of 60% via the chamber volume per minute using of a flowmeter. The livers were harvested and dissected for further analysis. To assess histological examinations, liver tissues were fixed in 10% buffered formalin and processed by a microtome. These sections were then stained with hematoxylin and eosin (H&E) before examination under a light microscope [28]. A four-point scoring system was employed to evaluate the severity of FLHS in the chicken livers, based on the presence and extent of acentric nuclei within hepatocytes. Acentric nuclei are displaced from their central position due to the accumulation of fat droplets within the cells. Scores were ranged from 1 (indicating normal liver with centrally located nuclei) to 4 (representing severe FLHS with extensive acentric nuclei), with 2 and 3 signifying mild and moderate degrees of acentricity, respectively (Fig 2).

## Statistical analysis

The study was conducted in a CRD using a 2 × 4 factorial layout. Data were subjected to the GLM procedures of SAS 9.4 software [29] according to the following model:

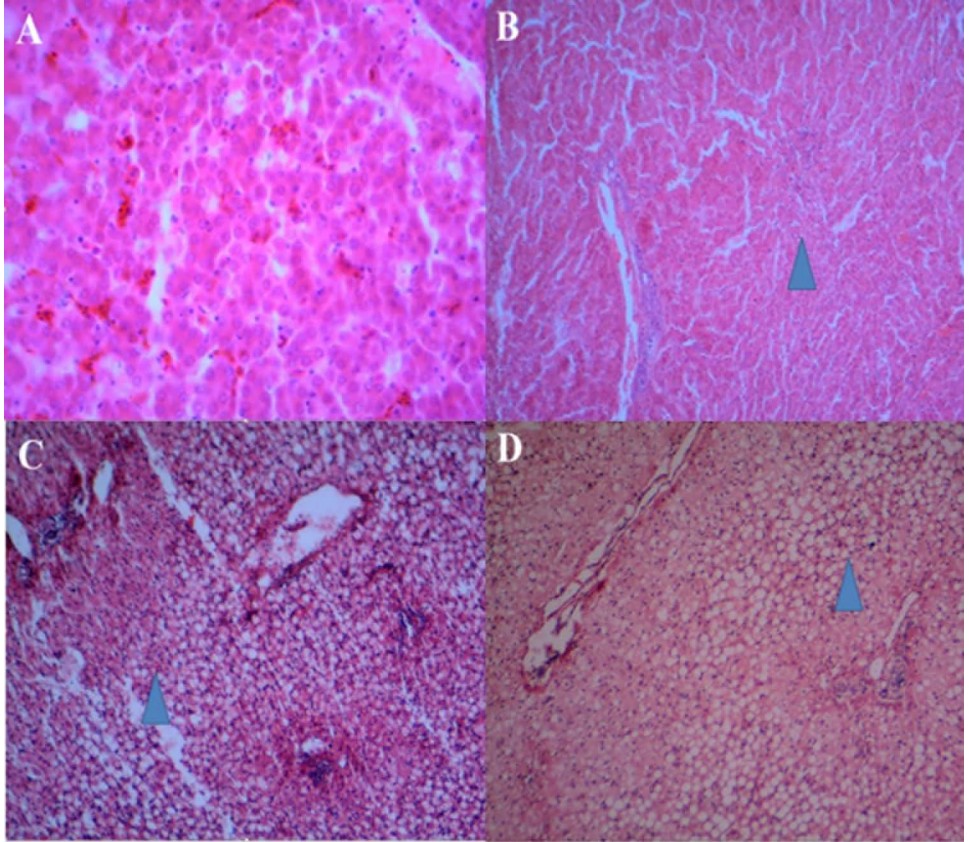

**Fig 2. Representative liver histopathology under H&E staining at increasing grades of fatty change.** (A) Normal liver shows uniform architecture with centrally located nuclei (score 1). (B) Mild fatty change (score 2) exhibits minimal fatty vacuoles (clear spaces) within hepatocytes. (C) Moderate fatty change (score 3) demonstrates increased cytoplasmic vacuolization. (D) Severe fatty change (score 4) displays extensive fatty infiltration obscuring most cytoplasm. Arrowhead highlights an acentric nucleus, a characteristic finding in steatosis.

$$Y_{ijk} = \mu + A_i + B_j + AB_{ij} + e_{ijk}$$

where $Y_{ijk}$ = observed value for a particular trait, $\mu$ = overall mean, $A_i$ = effect of the basal diet (SD and CD), $B_j$ = effect of the herbal mixture, $AB_{ij}$ = the respective interaction of ith and jth levels of dietary SDCD and herbal additives, and $e_{ijk}$ = random error associated with the ijkth recording. The treatment means were separated by Duncan's multiple-range test at P ≤ 0.05 statistical level. The effects of the main factors were not considered, whenever the interaction was significant. The non-parametric method of Kruskal-Wallis test was used to analyze liver histopathology scores [29].

## Results

### Laying performance

Tables 3–6 present the effects of dietary interventions on FI, EP, EW, and FCR of laying hens, respectively. Although no significant differences in FI were observed among treatments during the first four weeks of the experiment, both the main effects of the basal diets (SD vs. CD) and the herbal additives became significant in weeks 5 and 6 (Table 3). Hens fed the CD showed reduced FI compared to those on the SD (P = 0.0001). Among the herbal treatments, Mix 2 led to the highest FI during weeks 5 and 6 (P = 0.0053 and P = 0.0003, respectively). From weeks 7–12, a significant interaction between

**Table 3. Effect of different combinations of herbal additives with standard and challenge diet on the feed intake of laying hens during weeks 1 to 12 of the trial (65 to 77 weeks of age).**

| Treatments | Feed intake (g/hen/day) | | | | | | | | | | | |
|---|---|---|---|---|---|---|---|---|---|---|---|---|
| | Week1 | Week2 | Week3 | Week4 | Week5 | Week6 | Week7 | Week8 | Week9 | Week10 | Week11 | Week12 |
| **Main effects** | | | | | | | | | | | | |
| SD | 114.56 | 116.06 | 111.37 | 120.97 | 117.40[a] | 117.84[a] | 117.96[a] | 118.19[a] | 118.53[a] | 118.44[a] | 118.78[a] | 118.84[a] |
| CD | 114.40 | 116.40 | 110.69 | 118.62 | 105.34[b] | 103.62[b] | 102.84[b] | 103.31[b] | 96.87[b] | 96.87[b] | 96.87[b] | 96.87[b] |
| C | 113.44 | 114.31[b] | 110.75 | 117.00 | 110.44[b] | 108.81[b] | 107.19[c] | 107.62[c] | 103.50[c] | 103.31[c] | 104.00[c] | 104.12[b] |
| Mix 1 | 113.31 | 114.81[ab] | 110.31 | 119.62 | 109.50[b] | 109.50[b] | 109.50[bc] | 109.50[bc] | 104.37[c] | 104.37[bc] | 104.37[bc] | 104.37[b] |
| Mix 2 | 115.50 | 117.75[a] | 112.12 | 122.81 | 114.37[a] | 114.06[a] | 114.06[a] | 115.00[a] | 114.12[a] | 114.12[a] | 114.12[a] | 114.12[a] |
| Mix 3 | 115.69 | 118.06[a] | 110.94 | 119.75 | 111.19[b] | 110.56[b] | 110.87[b] | 110.87[b] | 108.81[b] | 108.81[b] | 108.81[b] | 108.81[b] |
| **SEM** | 12.01 | 20.13 | 22.21 | 55.17 | 15.20 | 11.81 | 17.36 | 18.07 | 37.24 | 39.97 | 40.07 | 40.23 |
| **Interactions** | | | | | | | | | | | | |
| SD | 113.13 | 112.13 | 110.88 | 116.25 | 115.13 | 116.88 | 117.38[a] | 118.25[a] | 117.13[a] | 116.75[a] | 118.13[a] | 118.38[a] |
| SD + Mix 1 | 113.75 | 116.63 | 110.38 | 123.25 | 116.50 | 116.50 | 116.50[a] | 116.50[a] | 116.88[a] | 116.88[a] | 116.88[a] | 116.88[a] |
| SD + Mix 2 | 115.00 | 116.88 | 112.38 | 121.75 | 119.38 | 119.38 | 119.38[a] | 119.38[a] | 120.00[a] | 120.00[a] | 120.00[a] | 120.00[a] |
| SD + Mix 3 | 116.38 | 118.63 | 111.88 | 122.63 | 118.63 | 118.63 | 118.63[a] | 118.63[a] | 120.13[a] | 120.13[a] | 120.13[a] | 120.13[a] |
| CD | 113.75 | 116.50 | 110.63 | 117.75 | 105.75 | 100.75 | 97.00[d] | 97.00[d] | 89.88[d] | 89.88[d] | 89.88[d] | 89.88[d] |
| CD + Mix 1 | 112.88 | 113.00 | 110.25 | 116.00 | 102.50 | 102.50 | 102.50[c] | 102.50[c] | 91.88 cd | 91.88 cd | 91.88 cd | 91.88 cd |
| CD + Mix 2 | 116.00 | 118.63 | 111.88 | 123.88 | 109.38 | 108.75 | 108.75[b] | 110.63[b] | 108.25[b] | 108.25[b] | 108.25[b] | 108.25[b] |
| CD + Mix 3 | 115.00 | 117.50 | 110.00 | 116.88 | 103.75 | 102.50 | 103.13[c] | 103.13[c] | 97.50[c] | 97.50[c] | 97.50[c] | 97.50[c] |
| **SEM** | 3.46 | 4.49 | 4.71 | 7.43 | 3.90 | 3.44 | 4.16 | 4.25 | 6.10 | 6.32 | 6.33 | 6.34 |
| **Significant levels** | | | | | | | | | | | | |
| SDCD | 0.8575 | 0.7604 | 0.5619 | 0.2121 | 0.0001 | 0.0001 | 0.0001 | 0.0001 | 0.0001 | 0.0001 | 0.0001 | 0.0001 |
| HA | 0.0985 | 0.0377 | 0.7306 | 0.1908 | 0.0053 | 0.0003 | 0.0002 | 0.0001 | 0.0001 | 0.0001 | 0.0001 | 0.0001 |
| SDCD*HA | 0.7264 | 0.0780 | 0.9496 | 0.1771 | 0.1202 | 0.0889 | 0.0153 | 0.0015 | 0.0036 | 0.0061 | 0.0034 | 0.0031 |

SD = standard diet; CD = challenge diet; C = control (without additive); Mix 1 = turmeric, fumitory, green tea and milk thistle powder; Mix 2 = lemon, black pepper, sumac and chicory powder; Mix 3 = garlic, artichoke, ginger and shallot powder; HA = herbal additive; SDCD*HA = interaction effect of dietary treatments.

a–d Means with no common superscript within each column are significantly (P ≤ 0.05) different.

SEM = standard error of the mean.

dietary treatments and FI was observed (P = 0.0153, P = 0.0015, P = 0.0036, P = 0.0061, P = 0.0034, and P = 0.0031, respectively). During this period, hens receiving the SD, with or without herbal additives, generally showed higher FI. Among the CD groups, CD + Mix 2 resulted in the highest FI and outperformed the other CD-based treatments (Table 3). The excel file of performance pure data has been presented as S2.

EP rate was not significantly affected by the experimental treatments during the first four weeks of the trial (P > 0.05) (Table 4). From week 5 onward, the interaction effects of SD and CD plus herbal mixes on EP were significant (P = 0.0001). Throughout the trial, the CD and CD with herbal mixes showed reduced EP compared to the SD and the SD with herbal mixes. The exception was the CD + Mix 2, which showed no significant difference from the SD group (P > 0.05).

No significant differences were observed in the EW of laying hens during the first four weeks (P > 0.05), except for a significant interaction effect in the first week and a main effect of herbal additives in week 3 (Table 5). In the first week, the group fed with SD had greater EW compared to the SD + Mix 1 (P = 0.0342). In the third week, Mix 2 resulted in a higher EW relative to the control group (P = 0.0445). In addition, the main effects of SD and CD were significant from weeks 5–12, with laying hens fed the CD having lower EW than those fed the SD (P = 0.0001). From weeks 5–9, the main effects

**Table 4. Effect of different combinations of herbal additives with standard and challenge diet on the egg production ratio (%) of laying hens during weeks 1 to 12 of the trial (65 to 77 weeks of age).**

| Treatments | Hen-day egg production (%) | | | | | | | | | | | |
|---|---|---|---|---|---|---|---|---|---|---|---|---|
| | Week1 | Week2 | Week3 | Week4 | Week5 | Week6 | Week7 | Week8 | Week9 | Week10 | Week11 | Week12 |
| **Main effects** | | | | | | | | | | | | |
| SD | 88.37 | 86.68 | 85.03 | 85.81 | 84.84[a] | 84.37[a] | 83.78[a] | 83.41[a] | 82.09[a] | 81.25[a] | 80.69[a] | 80.40[a] |
| CD | 88.58 | 86.86 | 85.05 | 85.68 | 75.59[b] | 75.50[b] | 75.50[b] | 75.53[b] | 71.66[b] | 71.28[b] | 70.34[b] | 70.34[b] |
| C | 88.44 | 87.20 | 85.53 | 86.42 | 80.87[b] | 79.87[b] | 79.75[b] | 79.25[b] | 75.12[bc] | 73.19[c] | 71.81[b] | 71.25[b] |
| Mix 1 | 88.67 | 86.69 | 85.03 | 85.61 | 78.56[c] | 78.56[b] | 78.31[bc] | 78.31[b] | 74.00[c] | 73.50[c] | 72.69[b] | 72.69[b] |
| Mix 2 | 88.00 | 86.64 | 84.84 | 85.64 | 83.31[a] | 83.25[a] | 82.87[a] | 82.81[a] | 80.69[a] | 80.69[a] | 80.31[a] | 80.31[a] |
| Mix 3 | 88.81 | 86.55 | 84.75 | 85.31 | 78.12[c] | 78.06[c] | 77.62[c] | 77.50[b] | 77.69[b] | 77.69[b] | 77.25[a] | 77.25[a] |
| **SEM** | 1.83 | 1.79 | 1.63 | 1.43 | 5.76 | 5.87 | 5.86 | 6.23 | 15.89 | 16.76 | 20.51 | 20.89 |
| **Interactions** | | | | | | | | | | | | |
| SD | 88.62 | 87.40 | 85.69 | 86.87 | 86.75[a] | 85.12[a] | 84.87[a] | 83.75[a] | 84.75[a] | 81.37[a] | 80.12[a] | 79.00[a] |
| SD + Mix 1 | 88.37 | 86.12 | 84.50 | 85.19 | 84.00[b] | 84.00[ab] | 83.50[a] | 83.50[a] | 81.12[a] | 81.12[a] | 80.50[a] | 80.50[a] |
| SD + Mix 2 | 87.75 | 86.31 | 85.06 | 85.94 | 84.25[ab] | 84.12[ab] | 83.37[a] | 83.25[a] | 81.00[a] | 81.00[a] | 80.75[a] | 80.75[a] |
| SD + Mix 3 | 88.75 | 86.87 | 84.87 | 85.26 | 84.37[ab] | 84.25[ab] | 83.37[a] | 83.12[a] | 81.50[a] | 81.50[a] | 81.37[a] | 81.37[a] |
| CD | 88.25 | 87.00 | 85.37 | 85.97 | 75.00[c] | 74.62[c] | 74.62[b] | 74.75[b] | 65.50[c] | 65.00[c] | 63.50[c] | 63.50[c] |
| CD + Mix 1 | 88.97 | 87.25 | 85.56 | 86.04 | 73.12 cd | 73.12 cd | 73.12[bc] | 73.12[bc] | 66.87[c] | 65.87[c] | 64.87[c] | 64.87[c] |
| CD + Mix 2 | 88.25 | 86.97 | 84.62 | 85.34 | 82.37[b] | 82.37[b] | 82.37[a] | 82.37[a] | 80.37[a] | 80.37[a] | 79.87[a] | 79.87[a] |
| CD + Mix 3 | 88.86 | 86.22 | 84.62 | 85.36 | 71.87[d] | 71.87[d] | 71.87[c] | 71.87[c] | 73.87[b] | 73.87[b] | 73.12[b] | 73.12[b] |
| **SEM** | 1.82 | 1.78 | 1.63 | 1.43 | 5.75 | 2.42 | 2.42 | 2.50 | 3.99 | 4.09 | 4.53 | 4.57 |
| **Significant levels** | | | | | | | | | | | | |
| SDCD | 0.5383 | 0.5835 | 0.9612 | 0.6477 | 0.0001 | 0.0001 | 0.0001 | 0.0001 | 0.0001 | 0.0001 | 0.0001 | 0.0001 |
| HA | 0.3576 | 0.5193 | 0.3221 | 0.0667 | 0.0001 | 0.0001 | 0.0001 | 0.0001 | 0.0001 | 0.0001 | 0.0001 | 0.0001 |
| SDCD*HA | 0.7334 | 0.1988 | 0.3151 | 0.1774 | 0.0001 | 0.0001 | 0.0001 | 0.0001 | 0.0001 | 0.0001 | 0.0001 | 0.0001 |

SD = standard diet; CD = challenge diet; C = control (without additive); Mix 1 = turmeric, fumitory, green tea and milk thistle powder; Mix 2 = lemon, black pepper, sumac and chicory powder; Mix 3 = garlic, artichoke, ginger and shallot powder; HA = herbal additive; SDCD*HA = interaction effect of dietary treatments.

a–d Means with no common superscript within each column are significantly (P ≤ 0.05) different.

SEM = standard error of the mean.

of herbal additives on EW were significant. Diets containing Mix 2 and Mix 3 consistently resulted in higher EW than the control diet (P = 0.0002, P = 0.0002, P = 0.0005, and P = 0.0044, respectively).

FCR was not affected by the experimental treatments between weeks 1–4 and weeks 9–12 (P > 0.05) (Table 6). The interaction effects of SD and CD with herbal additives were significant during weeks 5–8. In week 5, the lowest FCR was observed in the SD, SD + Mix 3, and CD + Mix 2, which were significantly different from CD, CD + Mix 1, and CD + Mix 3 (P = 0.0035). In week 6, the lowest FCR belonged to the CD + Mix 2 and SD diets, while the highest FCR was related to CD, CD + Mix 4, and CD + Mix 3 (P = 0.0048). In week 7, the CD + Mix 2, SD, SD + Mix 3, and CD groups had the lowest FCR and the CD + Mix 1 and CD + Mix 3 groups had the highest (P = 0.0239). Moreover, CD + Mix 2 and CD + Mix 3 had the lowest and highest FCR, respectively, when measured at week 8 (P = 0.0467).

## Hematological criteria

Tables 7–9 show hematological criteria of laying hens fed with herbal additives in both SD and CD at the beginning, middle, and end of the experimental period. Feeding the CD resulted in elevated levels of AST compared to the SD in the middle and at the end of the trial (P = 0.0001). The interaction effects of dietary treatments on AST enzyme levels

**Table 5. Effect of different combinations of herbal additives with standard and challenge diet on the egg weight (g) of laying hens during weeks 1 to 12 of the trial (65 to 77 weeks of age).**

| Treatments | Egg weight (g) | | | | | | | | | | | |
|---|---|---|---|---|---|---|---|---|---|---|---|---|
| | Week1 | Week2 | Week3 | Week4 | Week5 | Week6 | Week7 | Week8 | Week9 | Week10 | Week11 | Week12 |
| **Main effects** | | | | | | | | | | | | |
| SD | 62.56 | 63.62 | 63.17 | 64.69 | 64.37a | 64.37a | 64.43a | 64.39a | 64.51a | 64.56a | 64.65a | 64.75a |
| CD | 62.47 | 63.58 | 63.36 | 65.25 | 61.75b | 61.09b | 61.00b | 61.09b | 61.19b | 60.87b | 60.81b | 61.00b |
| C | 62.56 | 63.31 | 62.69b | 64.94 | 62.59b | 62.03b | 61.91b | 61.87b | 62.16b | 62.16b | 62.31 | 62.50 |
| Mix 1 | 62.31 | 63.42 | 63.37ab | 64.20 | 62.53b | 61.84b | 61.91b | 62.09b | 62.14b | 62.06b | 62.16 | 62.21 |
| Mix 2 | 62.62 | 63.81 | 63.81a | 65.49 | 63.44a | 63.43a | 63.44a | 63.41a | 63.37a | 63.06ab | 62.94 | 63.12 |
| Mix 3 | 62.56 | 63.87 | 63.19ab | 65.25 | 63.69a | 63.62a | 63.62a | 63.59a | 63.72a | 63.59a | 63.53 | 63.69 |
| **SEM** | 1.73 | 1.06 | 1.21 | 2.05 | 1.31 | 1.81 | 1.83 | 1.83 | 2.20 | 3.59 | 4.28 | 4.21 |
| **Interactions** | | | | | | | | | | | | |
| SD | 63.25a | 63.50 | 62.50 | 64.50 | 63.93 | 63.93 | 64.18 | 64.00 | 64.06 | 64.31 | 64.50 | 64.87 |
| SD + Mix 1 | 61.62b | 63.12 | 63.31 | 64.27 | 63.69 | 63.69 | 63.69 | 63.68 | 63.66 | 63.62 | 63.68 | 63.66 |
| SD + Mix 2 | 62.87ab | 63.87 | 64.00 | 65.11 | 64.50 | 64.50 | 64.50 | 64.43 | 64.87 | 64.87 | 64.87 | 64.87 |
| SD + Mix 3 | 62.50ab | 64.00 | 62.87 | 64.87 | 65.37 | 65.37 | 65.37 | 65.43 | 65.44 | 65.44 | 65.56 | 65.62 |
| CD | 61.87ab | 63.12 | 62.87 | 65.37 | 61.25 | 60.12 | 59.62 | 59.75 | 60.25 | 60.00 | 60.12 | 60.12 |
| CD + Mix 1 | 63.00ab | 63.71 | 63.44 | 64.12 | 61.37 | 60.00 | 60.12 | 60.50 | 60.62 | 60.50 | 60.62 | 60.75 |
| CD + Mix 2 | 62.37ab | 63.75 | 63.62 | 65.87 | 62.37 | 62.37 | 62.37 | 62.37 | 61.87 | 61.25 | 61.00 | 61.37 |
| CD + Mix 3 | 62.62ab | 63.75 | 63.50 | 65.62 | 62.00 | 61.87 | 61.87 | 61.75 | 62.00 | 61.75 | 61.50 | 61.75 |
| **SEM** | 1.31 | 1.03 | 1.10 | 1.43 | 1.14 | 1.34 | 1.33 | 1.35 | 1.48 | 1.89 | 2.07 | 2.05 |
| **Significant levels** | | | | | | | | | | | | |
| SDCD | 0.7766 | 0.8754 | 0.4983 | 0.1240 | 0.0001 | 0.0001 | 0.0001 | 0.0001 | 0.0001 | 0.0001 | 0.0001 | 0.0001 |
| HA | 0.9111 | 0.3239 | 0.0445 | 0.0722 | 0.0094 | 0.0002 | 0.0002 | 0.0005 | 0.0044 | 0.0752 | 0.2298 | 0.1865 |
| SDCD*HA | 0.0342 | 0.5568 | 0.6163 | 0.7240 | 0.4319 | 0.2669 | 0.0925 | 0.1424 | 0.8498 | 0.8509 | 0.8302 | 0.6418 |

SD = standard diet; CD = challenge diet; C = control (without additive); Mix 1 = turmeric, fumitory, green tea and milk thistle powder; Mix 2 = lemon, black pepper, sumac and chicory powder; Mix 3 = garlic, artichoke, ginger and shallot powder; HA = herbal additive; SDCD*HA = interaction effect of dietary treatments.

a–d Means with no common superscript within each column are significantly (P ≤ 0.05) different.

SEM = standard error of the mean.

in the middle and at the end of the trial were significant (P = 0.0001). Specifically, the lowest AST level was observed in laying hens fed with SD + Mix 3 and CD + Mix 2, while the highest levels were associated with the CD and CD + Mix 1 (Table 7).

The interaction effects of CD and SD with herbal additives on the concentration of TP were significant at both the middle and the end of the trial (Table 7). In the middle of the trial, hens fed the CD + Mix 2 had the highest TP concentration, while those on the CD and CD + Mix 1 showed the lowest (P = 0.0428). By the end of the trial, the highest TP concentrations were observed in the CD + Mix 2, SD + Mix 2, and SD + Mix 3 groups, whereas the SD, CD, and CD + Mix 1 groups had the lowest (P = 0.050).

For UA, the interaction effects of experimental treatments were significant at both the middle and the end of the experiment (Table 7). Overall, the SD without additives resulted in the highest blood UA levels. At mid-trial, this was significantly higher than those in the SD + Mix 2, SD + Mix 3, and CD + Mix 2 groups (P = 0.0006). By the end of the trial, the SD group also showed significantly higher UA concentrations compared to SD + Mix 1, SD + Mix 2, SD + Mix 3, and CD + Mix 2 (P = 0.0004). The lowest UA concentrations were consistently observed in the SD + Mix 2 and SD + Mix 3 groups.

**Table 6. Effect of different combinations of herbal additives with standard and challenge diet on the feed conversion ratio of laying hens during weeks 1 to 12 of the trial (65 to 77 weeks of age).**

| Treatments | Feed conversion ratio (g/g) | | | | | | | | | | | |
|---|---|---|---|---|---|---|---|---|---|---|---|---|
| | Week1 | Week2 | Week3 | Week4 | Week5 | Week6 | Week7 | Week8 | Week9 | Week10 | Week11 | Week12 |
| **Main effects** | | | | | | | | | | | | |
| SD | 2.07 | 2.10 | 2.07 | 2.17 | 2.15[b] | 2.17[b] | 2.19 | 2.20 | 2.23 | 2.26 | 2.27 | 2.28 |
| CD | 2.06 | 2.11 | 2.05 | 2.12 | 2.26[a] | 2.25[a] | 2.24 | 2.24 | 2.22 | 2.24 | 2.28 | 2.28 |
| C | 2.051 | 2.071 | 2.068 | 2.081 | 2.191 | 2.199[ab] | 2.168[b] | 2.187 | 2.226 | 2.277 | 2.331 | 2.350 |
| Mix 1 | 2.054 | 2.091 | 2.050 | 2.175 | 2.233 | 2.259[a] | 2.275[a] | 2.262 | 2.268 | 2.290 | 2.312 | 2.306 |
| Mix 2 | 2.096 | 2.130 | 2.062 | 2.193 | 2.164 | 2.160[b] | 2.168[b] | 2.187 | 2.229 | 2.241 | 2.250 | 2.250 |
| Mix 3 | 2.084 | 2.136 | 2.075 | 2.150 | 2.242 | 2.232[ab] | 2.262[ab] | 2.262 | 2.203 | 2.207 | 2.225 | 2.218 |
| **SEM** | 0.008 | 0.011 | 0.011 | 0.025 | 0.013 | 0.010 | 0.016 | 0.015 | 0.034 | 0.037 | 0.044 | 0.046 |
| **Interactions** | | | | | | | | | | | | |
| SD | 2.018 | 2.021 | 2.087 | 2.075 | 2.076[d] | 2.147 cd | 2.150[b] | 2.200[abc] | 2.158 | 2.236 | 2.275 | 2.312 |
| SD + Mix 1 | 2.090 | 2.145 | 2.050 | 2.250 | 2.178 cd | 2.178 cd | 2.212[ab] | 2.212[abc] | 2.263 | 2.264 | 2.275 | 2.275 |
| SD + Mix 2 | 2.085 | 2.120 | 2.050 | 2.175 | 2.197[bcd] | 2.200[bcd] | 2.225[ab] | 2.225[abc] | 2.283 | 2.283 | 2.287 | 2.287 |
| SD + Mix 3 | 2.100 | 2.134 | 2.100 | 2.212 | 2.151[d] | 2.154 cd | 2.187[b] | 2.187[bc] | 2.253 | 2.253 | 2.250 | 2.250 |
| CD | 2.084 | 2.121 | 2.050 | 2.087 | 2.306[ab] | 2.251[abc] | 2.187[b] | 2.175[bc] | 2.294 | 2.317 | 2.387 | 2.387 |
| CD + Mix 1 | 2.017 | 2.038 | 2.050 | 2.100 | 2.287[abc] | 2.341[a] | 2.337[a] | 2.312[ba] | 2.273 | 2.316 | 2.350 | 2.337 |
| CD + Mix 2 | 2.108 | 2.140 | 2.075 | 2.212 | 2.132[d] | 2.119[d] | 2.112[b] | 2.150[c] | 2.176 | 2.199 | 2.212 | 2.212 |
| CD + Mix 3 | 2.067 | 2.138 | 2.050 | 2.087 | 2.334[a] | 2.310[ab] | 2.337[a] | 2.337[a] | 2.153 | 2.161 | 2.200 | 2.187 |
| **SEM** | 0.090 | 0.106 | 0.107 | 0.158 | 0.115 | 0.104 | 0.128 | 0.125 | 0.186 | 0.193 | 0.210 | 0.215 |
| **Significant levels** | | | | | | | | | | | | |
| SDCD | 0.8502 | 0.8783 | 0.5623 | 0.1610 | 0.0002 | 0.0019 | 0.1258 | 0.2352 | 0.7386 | 0.8243 | 0.7674 | 1.000 |
| HA | 0.4149 | 0.2614 | 0.9237 | 0.2131 | 0.1978 | 0.0516 | 0.0286 | 0.1368 | 0.7966 | 0.6151 | 0.4384 | 0.3268 |
| SDCD*HA | 0.1536 | 0.0618 | 0.7448 | 0.2418 | 0.0035 | 0.0048 | 0.0239 | 0.0467 | 0.2226 | 0.4632 | 0.5175 | 0.6516 |

SD = standard diet; CD = challenge diet; C = control (without additive); Mix 1 = turmeric, fumitory, green tea and milk thistle powder; Mix 2 = lemon, black pepper, sumac and chicory powder; Mix 3 = garlic, artichoke, ginger and shallot powder; HA = herbal additive; SDCD*HA = interaction effect of dietary treatments.

a–d Means with no common superscript within each column are significantly (P ≤ 0.05) different.

SEM = standard error of the mean.

The main effects of SD and CD on TG concentration in the second sampling and their interaction effect in the third sampling were significant (Table 8). Based on the results, feeding CD to laying hens increased TG concentration when compared to the SD (P = 0.0001). Similarly, in the third sampling, birds fed with CD, CD + Mix 1, and CD + Mix 3 had the highest amount of TG, and SD + Mix 1 had the lowest (P = 0.0001).

In the second and third samplings, the interaction effects of dietary treatments on TC concentration were significant (Table 8). The highest amount of TC in the second sampling was observed in layers fed with CD + Mix 3, and the lowest in the diet of laying hens fed with SD + Mix 3 (P = 0.0001). In the third sampling, the highest concentration of TC resulted from feeding CD + Mix 1 and CD + Mix 3, and the lowest was observed by feeding SD + Mix 3 (P = 0.0001).

The effect of treatments on HDL concentration exhibited a statistically significant interaction at the second and third samplings (P = 0.0122 and P = 0.0028, respectively). The SD + Mix 3 group had the highest HDL concentration compared to the other treatments (Table 8). As for LDL, the main effect of the SD and CD was significant in the second sampling, with CD increased the concentration of LDL compared to the SD (P = 0.0001) (Table 8).

The interaction effects of diets showed a significant effect on blood Hb concentration and heterophile and lymphocyte count in both the second and third samplings (Table 9). In the second sampling, CD + Mix 2 group had the highest Hb

**Table 7. Effect of different combinations of herbal additives with standard and challenge diet on the blood parameters of laying hens (Start, Middle, End).**

| Treatments | AST(U/L) | | | TP(g/dL) | | | UA(mg/dL) | | |
|---|---|---|---|---|---|---|---|---|---|
| | Start (wk 65) | Middle (wk 70) | End (wk 77) | Start (wk 65) | Middle (wk 70) | End (wk 77) | Start (wk 65) | Middle (wk 70) | End (wk 77) |
| **Main effects** | | | | | | | | | |
| SD | 213.57 | 224.78[b] | 232.97[b] | 6.41 | 6.79[a] | 6.90[a] | 6.18 | 6.19[b] | 6.56[b] |
| CD | 213.55 | 259.84[a] | 295.50[a] | 6.19 | 6.24[b] | 6.44[b] | 6.30 | 6.53[a] | 7.28[a] |
| C | 214.13 | 253.62[a] | 288.56[a] | 6.21 | 5.97[b] | 5.99[c] | 6.19 | 6.79[a] | 7.56[a] |
| Mix 1 | 212.78 | 253.31[a] | 279.31[a] | 6.35 | 6.42[b] | 6.53[b] | 6.19 | 6.71[a] | 7.14[b] |
| Mix 2 | 214.59 | 220.31[b] | 232.06[c] | 6.28 | 7.28[a] | 7.40[a] | 6.26 | 5.91[b] | 6.33[c] |
| Mix 3 | 212.75 | 242.00[a] | 257.00[b] | 6.37 | 6.40[b] | 6.77[b] | 6.32 | 6.03[b] | 6.65[c] |
| **SEM** | 70.06 | 469.77 | 531.16 | 0.279 | 0.559 | 0.430 | 0.122 | 0.262 | 0.332 |
| **Interactions** | | | | | | | | | |
| SD | 214.25 | 234.75[b] | 253.87[b] | 6.37 | 6.35[bcd] | 6.12[c] | 6.12 | 6.89[a] | 7.69[a] |
| SD + Mix 1 | 209.46 | 232.87[b] | 235.87[bc] | 6.70 | 7.02[ab] | 7.06[ab] | 6.17 | 6.80[a] | 6.91[b] |
| SD + Mix 2 | 218.94 | 225.62[bc] | 239.37[bc] | 6.46 | 7.12[ab] | 7.30[a] | 6.31 | 5.62[c] | 5.72[c] |
| SD + Mix 3 | 211.65 | 205.87[c] | 202.75[d] | 6.12 | 6.68[abc] | 7.12[a] | 6.13 | 5.44[c] | 5.94[c] |
| CD | 214.01 | 272.50[a] | 323.25[a] | 6.05 | 5.60[d] | 5.85[c] | 6.26 | 6.70[ab] | 7.43[ab] |
| CD + Mix 1 | 216.10 | 273.75[a] | 322.75[a] | 6.00 | 5.81[d] | 6.00[c] | 6.21 | 6.62[ab] | 7.37[ab] |
| CD + Mix 2 | 210.23 | 215.00[bc] | 224.75 cd | 6.10 | 7.43[a] | 7.50[a] | 6.21 | 6.19[b] | 6.94[b] |
| CD + Mix 3 | 213.86 | 278.12[a] | 311.25[a] | 6.61 | 6.12 cd | 6.41[bc] | 6.52 | 6.62[ab] | 7.37[ab] |
| **SEM** | 8.37 | 21.67 | 23.05 | 0.528 | 0.747 | 0.656 | 0.350 | 0.513 | 0.576 |
| **Significant levels** | | | | | | | | | |
| SDCD | 0.9917 | 0.0001 | 0.0001 | 0.0943 | 0.0046 | 0.0065 | 0.1853 | 0.0090 | 0.0001 |
| HA | 0.8956 | 0.0001 | 0.0001 | 0.8325 | 0.0001 | 0.0001 | 0.6854 | 0.0001 | 0.0001 |
| SDCD*HA | 0.0799 | 0.0001 | 0.0001 | 0.0870 | 0.0428 | 0.0500 | 0.2647 | 0.0006 | 0.0004 |

SD = standard diet; CD = challenge diet; C = control (without additive); Mix 1 = turmeric, fumitory, green tea and milk thistle powder; Mix 2 = lemon, black pepper, sumac and chicory powder; Mix 3 = garlic, artichoke, ginger and shallot powder; HA = herbal additive; SDCD*HA = interaction effect of dietary treatments; AST = aspartate aminotransferase; TP = total protein; UA = uric acid.

a–d Means with no common superscript within each column are significantly (P ≤ 0.05) different.

SEM = standard error of the mean.

concentration, while the lowest values were observed in SD + Mix 2 and SD + Mix 3 (P = 0.0001). In the third sampling, the highest Hb levels were recorded in the CD and CD + Mix 1 groups, while SD + Mix 2 and SD + Mix 3 showed the lowest concentrations (P = 0.0002). Regarding the number of heterophils in the second and third samplings, the highest count was seen in the SD + Mix 1 group, while the lowest was found in SD + Mix 2 and SD + Mix 3 (P = 0.0001). In the second sampling, the highest lymphocyte counts were recorded in the SD, SD + Mix 1, and all CD groups, whereas the lowest was observed in SD + Mix 2 (P = 0.0108). In the third sampling, CD + Mix 1 exhibited the highest lymphocyte count, while SD + Mix 2 showed the lowest (P = 0.0030).

## Hepatic histopathology

Fig 3 illustrates the effects of various herbal mixes on the frequency of liver histopathology scores in laying hens. Statistical analysis revealed a significant difference (P = 0.002) among the treatment groups. Notably, hens fed the SD + Mix 2 and CD + Mix 2 exhibited significantly lower liver scores compared to those fed CD without additive. No significant differences were observed among the other experimental treatments (P > 0.05).

**Table 8. Effect of different combinations of herbal additives with standard and challenge diet on the blood parameters of laying hens (Start, Middle, End).**

| Treatments | TG (mg/dL) | | | TC (mg/dL) | | | HDL (mg/dL) | | | LDL (mg/dL) | | |
|---|---|---|---|---|---|---|---|---|---|---|---|---|
| | Start (wk 65) | Middle (wk 70) | End (wk 77) | Start (wk 65) | Middle (wk 70) | End (wk 77) | Start (wk 65) | Middle (wk 70) | End (wk 77) | Start (wk 65) | Middle (wk 70) | End (wk 77) |
| **Main effects** | | | | | | | | | | | | |
| SD | 1304.28 | 1410.63b | 1485.47b | 210.29 | 199.70b | 200.00b | 5.39 | 6.35 | 6.61 | 46.42 | 52.03b | 55.88 |
| CD | 1434.28 | 1580.88a | 2611.97a | 214.59 | 239.71a | 275.19a | 5.71 | 6.51 | 6.25 | 46.63 | 54.44a | 56.21 |
| C | 1382.19 | 1503.31 | 2257.8a | 213.64 | 230.12a | 255.42a | 5.07 | 5.70b | 5.30c | 46.15b | 52.88 | 57.10 |
| Mix 1 | 1373.44 | 1473.75 | 2195.8a | 215.12 | 233.00a | 258.62a | 5.41 | 5.91b | 5.91bc | 46.29ab | 53.88 | 55.59 |
| Mix 2 | 1388.25 | 1518.75 | 1682.9b | 209.71 | 221.47a | 230.75b | 5.87 | 6.59ab | 6.58b | 47.11a | 53.41 | 56.25 |
| Mix 3 | 1333.00 | 1487.19 | 2058.5a | 211.30 | 194.25b | 205.56c | 5.85 | 7.53a | 7.93a | 46.56ab | 52.78 | 55.25 |
| **SEM** | 127.64 | 272.29 | 120.5 | 108.43 | 423.49 | 531.16 | 2.43 | 2.12 | 1.60 | 1.38 | 3.31 | 9.01 |
| **Interactions** | | | | | | | | | | | | |
| SD | 1305.50 | 1462.50 | 1518.87bc | 211.00 | 225.12b | 235.87c | 5.01 | 5.15b | 5.11c | 45.90 | 52.50 | 58.37 |
| SD + Mix 1 | 1327.12 | 1371.87 | 1397.75c | 211.50 | 231.62ab | 233.37c | 4.63 | 5.32b | 5.56bc | 46.43 | 52.12 | 55.27 |
| SD + Mix 2 | 1338.37 | 1453.12 | 1564.50bc | 208.93 | 204.19c | 203.37d | 5.60 | 6.43b | 6.60b | 47.15 | 51.93 | 55.75 |
| SD + Mix 3 | 1245.62 | 1355.00 | 1460.75bc | 209.75 | 137.87d | 127.37e | 6.32 | 8.50a | 9.18a | 46.22 | 51.56 | 54.12 |
| CD | 1458.87 | 1544.12 | 2996.62a | 216.27 | 235.12ab | 275.00ab | 5.14 | 6.25b | 5.49bc | 46.41 | 53.26 | 55.83 |
| CD + Mix 1 | 1419.75 | 1575.62 | 2993.75a | 218.75 | 234.37ab | 283.87a | 6.19 | 6.50b | 6.26bc | 46.13 | 55.63 | 55.90 |
| CD + Mix 2 | 1438.12 | 1584.37 | 1801.25b | 210.49 | 238.75ab | 258.12b | 6.15 | 6.75b | 6.56b | 47.07 | 54.87 | 56.75 |
| CD + Mix 3 | 1420.37 | 1619.37 | 2656.25a | 212.86 | 250.62a | 283.75a | 5.37 | 6.56b | 6.68b | 46.90 | 54.00 | 56.37 |
| **SEM** | 113.00 | 165.20 | 346.79 | 10.41 | 20.58 | 16.86 | 1.56 | 1.45 | 1.26 | 1.17 | 1.82 | 3.00 |
| **Significant levels** | | | | | | | | | | | | |
| SDCD | 0.0823 | 0.0001 | 0.0001 | 0.1044 | 0.0001 | 0.0001 | 0.4174 | 0.6572 | 0.2548 | 0.4916 | 0.0001 | 0.6576 |
| HA | 0.5119 | 0.8797 | 0.0001 | 0.4703 | 0.0001 | 0.0001 | 0.4163 | 0.0033 | 0.0001 | 0.1122 | 0.2993 | 0.3221 |
| SDCD*HA | 0.6787 | 0.4248 | 0.0001 | 0.8758 | 0.0001 | 0.0001 | 0.1651 | 0.0122 | 0.0028 | 0.5998 | 0.1785 | 0.1501 |

SD = standard diet; CD = challenge diet; C = control (without additive); Mix 1 = turmeric, fumitory, green tea and milk thistle powder; Mix 2 = lemon, black pepper, sumac and chicory powder; Mix 3 = garlic, artichoke, ginger and shallot powder; HA = herbal additive; SDCD*HA = interaction effect of dietary treatments; TG = triglyceride; TC = total cholesterol; HDL = high density lipoprotein; LDL = Low density lipoprotein.

a–d Means with no common superscript within each column are significantly (P ≤ 0.05) different.

SEM = standard error of the mean.

## Discussion

This study aimed to investigate the effects of various medicinal plant mixtures on laying hen performance, blood chemistry, and liver health. These mixtures were administered alongside either a SD or a CD. Hens received a CD supplemented with Mix 2 (comprising lemon, black pepper, sumac, and chicory) demonstrated significant improvements in EP and FCR. Meanwhile, the adverse effects of CD on liver health have been mitigated. The chemical composition of the herbs in these mixes exhibited that a substantial portion of the identified compounds were phthalates and terpenes. Phthalides, a class of secondary metabolites, possess a broad spectrum of pharmacological activities, including anti-inflammatory and antimicrobial activities [30]. Terpens have well-known anti-inflammatory and antioxidant effects [31]. Terpens included caryophyllene, beta-bisabolene, limonene, and α-copaene, among others. Additionally, the presence of organosulfur compounds, another class of potentially beneficial natural products, was observed in garlic and shallot [16]. Notably, curcumin, a curcuminoid with established health benefits, constituted the predominant component in turmeric essential oil [19].

In this study, the CD was formulated with high energy and low protein content to induce FLHS [32]. Biologically, a high intake of energy relative to protein causes an imbalance in macronutrient metabolism, favoring lipogenesis over protein

**Table 9. Effect of different combinations of herbal additives with standard and challenge diet on the blood parameters of laying hens (Start, Middle, End).**

| Treatments | Hb(g/dL) | | | Heterophile | | | Lymphocyte | | |
|---|---|---|---|---|---|---|---|---|---|
| | Start (wk 65) | Middle (wk 70) | End (wk 77) | Start (wk 65) | Middle (wk 70) | End (wk 77) | Start (wk 65) | Middle (wk 70) | End (wk 77) |
| **Main effects** | | | | | | | | | |
| SD | 19.41 | 19.72[b] | 20.69[b] | 25.84 | 26.54 | 27.77 | 73.03 | 79.95[b] | 82.36[b] |
| CD | 18.67 | 24.09[a] | 28.59[a] | 25.63 | 26.40 | 27.76 | 72.83 | 83.23[a] | 90.09[a] |
| C | 19.00 | 23.27[a] | 27.42[a] | 25.41 | 27.03[b] | 29.37[a] | 72.70 | 84.09[a] | 90.62[a] |
| Mix 1 | 18.94 | 23.18[a] | 27.34[a] | 26.05 | 28.50[a] | 29.94[a] | 72.53 | 83.77[ab] | 91.65[a] |
| Mix 2 | 17.61 | 21.27[b] | 21.31[b] | 25.64 | 25.25[c] | 26.24[b] | 72.93 | 77.58[c] | 77.62[c] |
| Mix 3 | 20.61 | 19.91[b] | 22.50[b] | 25.83 | 25.12[c] | 25.50[b] | 73.56 | 80.94[b] | 85.00[b] |
| **SEM** | 6.054 | 4.99 | 10.507 | 0.718 | 2.133 | 5.16 | 1.69 | 16.60 | 22.89 |
| **Interactions** | | | | | | | | | |
| SD | 21.00 | 23.56[b] | 25.60[bc] | 25.65 | 27.56[b] | 30.44[ab] | 73.00 | 84.37[a] | 88.12[b] |
| SD+Mix 1 | 18.61 | 22.74[b] | 25.17[c] | 26.11 | 30.50[a] | 31.51[a] | 73.25 | 83.54[a] | 89.81[ab] |
| SD+Mix 2 | 17.30 | 16.28[c] | 15.74[d] | 25.92 | 24.50[c] | 24.22[d] | 72.77 | 73.41[c] | 74.37[d] |
| SD+Mix 3 | 20.75 | 16.31[c] | 16.25[d] | 25.67 | 23.62[c] | 24.87[d] | 73.09 | 78.50[b] | 77.12 cd |
| CD | 17.01 | 22.99[b] | 29.25[a] | 25.17 | 26.50[b] | 28.31[bc] | 72.40 | 83.81[a] | 93.12[ab] |
| CD+Mix 1 | 19.26 | 23.62[b] | 29.50[a] | 25.99 | 26.50[b] | 28.37[bc] | 71.81 | 84.00[a] | 93.50[a] |
| CD+Mix 2 | 17.92 | 26.25[a] | 26.87[abc] | 25.36 | 26.00[b] | 28.25[bc] | 73.09 | 81.75[ab] | 80.87[c] |
| CD+Mix 3 | 20.47 | 23.50[b] | 28.75[ab] | 25.99 | 26.62[b] | 26.12 cd | 74.02 | 83.37[a] | 92.87[ab] |
| **SEM** | 2.46 | 2.21 | 3.24 | 0.847 | 1.46 | 2.27 | 1.30 | 4.07 | 4.78 |
| **Significant levels** | | | | | | | | | |
| SDCD | 0.2298 | 0.0001 | 0.0001 | 0.3201 | 0.7016 | 0.9956 | 0.5429 | 0.0021 | 0.0001 |
| HA | 0.0721 | 0.0001 | 0.0001 | 0.1904 | 0.0001 | 0.0001 | 0.1391 | 0.0001 | 0.0001 |
| sSDCD*HA | 0.0923 | 0.0001 | 0.0002 | 0.4563 | 0.0001 | 0.0001 | 0.0632 | 0.0108 | 0.0030 |

SD = standard diet; CD = challenge diet; C = control (without additive); Mix 1 = turmeric, fumitory, green tea and milk thistle powder; Mix 2 = lemon, black pepper, sumac and chicory powder; Mix 3 = garlic, artichoke, ginger and shallot powder; HA = herbal additive; SDCD*HA = interaction effect of dietary treatments; Hb = hemoglobin.

a–d Means with no common superscript within each column are significantly (P ≤ 0.05) different.

SEM = standard error of the mean.

synthesis [33]. This results in excessive fat deposition, particularly in the liver, which impairs liver function and overall metabolic homeostasis [33].

Feeding the CD significantly decreased FI, EW, and EP rate from week 5 onward, with a concomitant increase in FCR compared to feeding a SD. These reductions in performance can be logically attributed to the metabolic disturbances caused by the CD's high-energy and low-protein contents [34]. Specifically, the accumulation of lipids in the liver and adipose tissue leads to an increased secretion of leptin—a hormone known to suppress appetite—thereby reducing FI [32,35]. Reduced nutrient intake *per se* contributes to a decline in EP and EW, as laying hens prioritize survival over reproduction under physiological stress. Moreover, impaired liver function can reduce the synthesis of essential compounds for egg formation, such as yolk precursors and lipoproteins, further compromising productivity [36]. Similar to our findings, a previous study documented a decline in EW, EP rate, and FI in hens fed a CD starting from weeks 2, 3, and 4, respectively [32]. In another, laying hens on a CD consumed less feed and had a lower EP than hens on a control diet [7]. The metabolic consequences of the CD extend beyond FI and egg output. Lipid accumulation in the liver is a precursor of FLHS, which physically damages hepatocytes and leads to systemic inflammation [37]. Inflammatory responses and

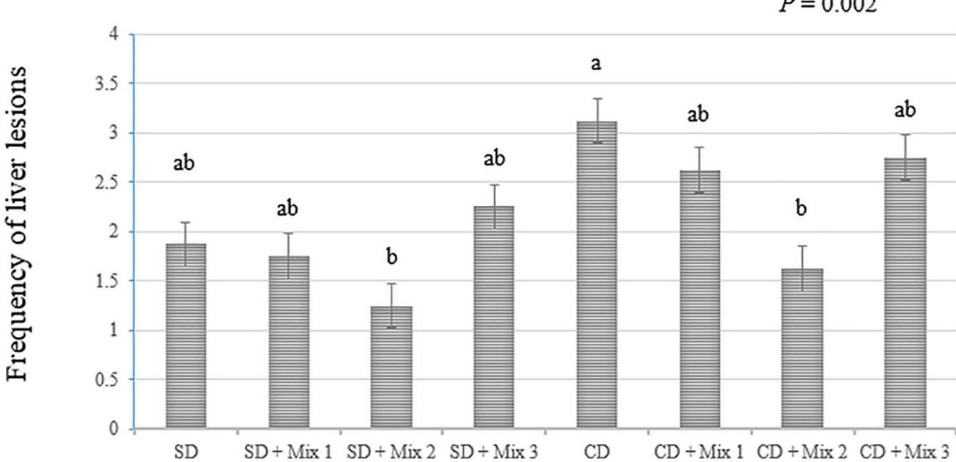

**Fig 3. The Effect of different combinations of herbal additives with standard and challenge diet on the liver histopathology score of laying hens.** SD, standard diet; CD, challenge diet; Mix 1 = turmeric, fumitory, green tea and milk thistle powder; Mix 2 = lemon, black pepper, sumac and chicory powder; Mix 3 = garlic, artichoke, ginger and shallot powder. Different letters indicate significant difference among different treatments at the 0.05 level.

oxidative stress disrupt hormonal regulation and blood flow to reproductive organs, which further suppressing productive performance [4,7,37,38]. In the current study, Mix 2 helped to prevent hepatic damage to a great extent in hens fed with the CD, which was associated with improved FI, EP, and FCR. This improvement is presumably due to the combined antioxidant, anti-inflammatory, and hepatoprotective actions of plant secondary metabolites in Mix 2. Phenolic compounds in sumac and chicory are known to scavenge reactive oxygen species and modulate lipid metabolism, thereby reducing hepatic steatosis [11,14]. Black pepper possesses piperine alkaloids, which have anti-inflammatory property and can enhance nutrient absorption [13]. Lemon contains flavonoids that can stabilize hepatic membranes and improve lipid profiles [12,39].

Two theories have been proposed for the pathogenesis of FLHS. The first theory suggests that FLHS begins with the excessive deposition of TG in the liver, disrupting the homeostasis of lipid metabolism [40]. The second theory proposes that oxidative stress in FLHS induces excessive reactive oxygen species in the liver, leading to the destruction of hepatic connective tissues, called reticulin [40]. Modulating these two processes (lipid accumulation and oxidative stress) through Mix 2 may have targeted both pathways of FLHS progression, leading to improved liver histopathology and overall health of laying hens. The specific combination in Mix 2 may have enhanced bioavailability or complementary mechanisms—such as simultaneous modulation of gut health, hepatic function, and systemic inflammation [11–14,22]. This synergy could explain why improvements were more consistent and robust in the CD + Mix 2 group.

Limited information is available about the role of medicinal plants in preventing FLHS in laying hens. A previous study found that adding a mixture of chicory, turmeric, artichoke, and milk thistle powder to a high energy/low protein diet did not affect EP or FCR in laying hens [41]. However, artichoke and thistle powder reduced FI, while turmeric powder increased EW compared to the control group [41]. Another study observed that broiler chickens fed diets containing chicory root powder showed significant improvement in growth performance, likely due to increased feed assimilation through changes in jejunum histomorphometry [42]. In one study, supplementation with 0.5% sumac in feed was found to slightly increase EW, although FI, FCR, and EP were not significantly affected [43]. Research has shown that adding lemon essential oil to quail's feed improved FCR and FI, possibly due to the active compounds and the antioxidant and antimicrobial properties of the oil [44]. A recent study demonstrated that the inclusion of garlic into laying hen diets could increase EW [45]. Another study reported increased EP, EW, and FI in groups treated with inulin [46]. Inconsistencies observed across these

studies may be attributed to factors such as variations in dosage, limited absorption of the additives, and potential interactions between different herbal plants.

Regardless of feed additives, laying hens fed with the CD exhibited higher blood AST levels compared to those on the SD at the middle and the end of the experiment. This elevation suggests hepatocellular injury, as AST is a key intracellular enzyme released into circulation when liver cells are damaged or inflamed [47]. The high energy and low protein contents of the CD may have promoted an excessive accumulation of lipids in the liver, which disrupts liver architecture, induces oxidative stress, and compromises hepatocyte integrity. These changes account for increased AST level in the bloodstream [32,48]. Notably, the inclusion of Mix 2 in the CD-fed group has counterbalanced AST level, indicating a potential protective effect of its herbal constituents on liver health. Botanical mixtures rich in polyphenols, flavonoids, and inulin can (1) quench reactive oxygen species, (2) stabilize hepatocyte membranes, (3) enhance phase-II detoxification, and (4) divert lipid flux away from *de novo* lipogenesis toward β-oxidation [11,13,49]. This supports the hypothesis that phytogenic additives can attenuate liver injury under metabolic stress conditions [22]. Our findings align with previous reports indicating that high-energy, low-protein diets are associated with increased AST levels in laying hens, reflecting diet-induced hepatic stress and early stages of fatty liver development [32,48]. Although direct studies on the effects of medicinal plants on FLHS in laying hens are scarce, available research in avian species suggests beneficial outcomes. For instance, lemon essential oil supplementation in quail diets has shown to reduce AST levels, potentially due to cytoprotective and antioxidative actions [44]. Some researchers have found that sumac supplementation had no significant effect on AST concentrations [50,51]. A combination of chicory, turmeric, artichoke, and milk thistle could not reduce AST levels when included in a challenge diet for laying hens [41]. In contrast, other studies have demonstrated that herbal blends containing milk thistle significantly lowered AST in hens with fatty liver, highlighting the hepatoprotective role of specific plant constituents [52]. These contrasting results may be attributed to various factors, including the dosage of herbal additives, interactions among different herbs, or the quality of the herbs used.

The use of herbal formulations in the SD increased blood TP concentration compared to the SD control. An elevation in TP may reflect improved dietary protein utilization and metabolic efficiency, potentially mediated by enhanced gut integrity, nutrient absorption, and systemic antioxidant capacity [53]. Herbal bioactive compounds—such as phenolics—are known to modulate gut microbiota, reduce oxidative damage to enterocytes, and improve intestinal morphology, thereby supporting a better nutrient digestion and absorption [53]. For example, ginger supplementation at a rate of 5 g/kg of diet has been reported to increase serum TP concentrations in broilers, likely due to an enhanced antioxidant status [54]. In addition, Oleforuh-Okoleh et al. [55] demonstrated that adding garlic to broiler diets increased blood TP compared to the control group. The only CD treatment with a TP concentration similar to the SD groups was CD + Mix 2. This finding suggests that Mix 2 may have improved protein retention or minimized catabolism despite the protein-deficient nature of the CD. A well-established relationship exists between dietary protein levels and plasma protein concentration [56]. Reduced TP levels in other CD-fed groups might result from impaired protein synthesis, increased protein turnover, or hepatic dysfunction commonly associated with fatty liver development [57,58]. In support of this statement, lemon essential oil has previously been shown to increase blood TP in quail, possibly through its antioxidant properties and positive impact on liver and gut function [44].

An increase in blood UA concentration observed in the CD groups was not mitigated by the addition of herbal supplements. The lowest UA concentrations were observed in the SD + Mix 2 and SD + Mix 3 groups. In birds, UA is a by-product of protein catabolism, which is primarily eliminated from the body in droppings. Previous research suggests that fatty liver may elevate circulatory UA levels, and UA has been proposed as an independent predictor of fatty liver risk [59]. In agreement with our results, one study reported that supplementing a high-energy, low-protein diet with chicory, turmeric, artichoke, and thistle powder did not significantly affect UA concentrations in laying hens [41]. Similarly, Mosayyeb Zadeh et al. [51] found that dietary inclusion of sumac 0.25, and 0.5% had no significant effect on UA levels in laying hens.

In the current study, blood TG concentration was significantly increased in hens fed with the CD at the second sampling point, reflecting disrupted lipid metabolism and hepatic lipid overload. Excess dietary energy and insufficient protein supply may impair hepatic lipid handling, causing TG to accumulate in the liver and leak into circulation. However, at the third sampling, the CD+Mix 2 attenuated such increase compared to other CD treatments. This suggests that Mix 2 may enhance lipid metabolism, reduce hepatic fat accumulation, or improve lipid clearance from the bloodstream. The lipid-lowering effect is likely mediated through the antioxidant and fat-modulatory properties of its herbal components. Studies suggest that sumac and its polyphenols may reduce TG levels by hindering fat digestion and absorption [51,60]. Similarly, an herbal mix containing lemon was reported to reduce TG levels in overweight rats [61]. This effect may be attributed to lemon's vitamin C and flavonoid content, which act as antioxidants [49,62].

TC concentration was also increased by feeding the CD at the second and third samplings, and the herbal additives were unable to restore this effect. An elevated TC is commonly associated with FLHS, as impaired lipid metabolism and liver dysfunction can lead to increased cholesterol synthesis and impaired excretion [6,48]. The inability of herbal mixtures to reduce TC under CD conditions may be due to insufficient duration of treatment, suboptimal dosing, or complex interactions among the bioactive compounds that diminished their cholesterol-lowering efficacy. The lowest TC concentration was observed in the SD+Mix 3 group which includes garlic and shallot—members of the *Allium* genus well-documented for their hypocholesterolemic effects. These effects are often attributed to sulfur-containing compounds that influence lipid metabolism and bile acid synthesis [16,63,64]. Akbarian et al. [65] showed a decrease in blood TC in laying hens fed ginger root powder (0.25%, 0.5%, and 0.75%) over an 8-week period. One possible mechanism involves the modulation of bile acids [54]. Artichoke, also found in Mix 3, may contribute by enhancing biliary excretion of cholesterol and supporting liver detoxification [66].

In the present study, feeding the CD increased blood LDL concentrations in laying hens compared to the SD at the second sampling. LDL, commonly referred to as "bad cholesterol," is a key contributor to lipid accumulation in blood vessels and a known risk factor for cardiovascular disease [67]. Despite this initial increase, supplementation with herbal additives in both SD and CD groups did not lead to significant changes in LDL levels, suggesting limited impact of the additives on LDL metabolism within the timeframe of the experiment. Overall, no reduction in blood HDL was observed in the CD groups, regardless of herbal additive inclusion, when compared to the SD group. HDL is a type of lipoprotein responsible for transporting cholesterol from peripheral tissues to the liver for excretion. It is often referred to as "good cholesterol" due to its positive impact on reducing the risk of cardiovascular diseases [68,69]. Several factors—including the severity and duration of fatty liver, dietary composition, nutritional status, and genetic predisposition—can influence HDL levels. In the present study, hens in the SD+Mix 3 group exhibited the highest HDL concentrations at both the second and third samplings compared to the other treatments. Mix 3 includes garlic, ginger, artichoke, and shallot—herbs that are rich in bioactive compounds like allicin, gingerols, and cynarin, which have been associated with improved lipid profiles through mechanisms such as enhanced bile secretion, reduced hepatic lipogenesis, and increased antioxidant capacity [16,53]. Previous research has shown that HDL cholesterol levels increased in laying quails fed diets containing 1%, 2%, and 4% garlic powder [70]. Likewise, Saki et al. [71] reported significantly higher serum HDL levels in laying hens fed diets supplemented with 8 or 12 g/kg of an herbal mixture. Moreover, an increase in serum HDL cholesterol was reported in all treated groups with ginger rhizome oil in Japanese quails, relative to the control group [72].

The concentration of Hb at different sampling points did not follow a consistent trend. Overall, Hb levels were elevated in laying hens fed the CD, irrespective of herbal additives, compared to those in the SD+Mix 2 and SD+Mix 3 groups. Previous studies have shown that a high-fat, low-protein diet increases Hb concentrations in both mice and laying hens compared to the control diet supplemented with herbal additives [61,73]. It is important to note that the relationship between fatty liver and Hb concentration in laying hens is complex and may be influenced by various factors, including the severity and etiology of fatty liver, as well as metabolic pathways involved in erythropoietin production and red blood cell turnover. Another study reported elevated circulating Hb levels in patients with fatty liver, with the increase correlating with

the degree of hepatic steatosis. In such cases, Hb may function as an antioxidant, mitigating some of the negative effects associated with the disease [74].

In the present study, no significant differences were observed in heterophile counts among hens fed the CD with or without herbal additives. However, the SD + Mix 1 group exhibited the highest heterophile counts at the second and third samplings. Lymphocyte numbers increased in the CD groups, while the lowest counts were observed in the SD + Mix 2 and SD + Mix 3 groups. According to a previous study, a low white blood cell count may suggest either a disease-free condition or suppressed bone marrow activity [75]. It is presumed that the medicinal plants included in Mix 2 and Mix 3 possess anti-inflammatory, antimicrobial, and immune system-stimulating properties. The potential alleviation of fatty liver symptoms may have contributed to the observed reduction in lymphocyte numbers. In support of this, one study found that dietary supplementation with 25 and 50 mg/kg garlic in broilers had no significant effect on lymphocyte counts, though the higher dose (50 mg/kg) significantly reduced heterophile counts compared to the control [76]. In contrast to our results, another study reported that garlic increased heterophile, lymphocyte, and the heterophile to lymphocyte ratio. The researchers attributed these responses to the stimulatory effect of garlic on the immune system [77]. In our study, the CD alone led to a higher lymphocyte count than CD + Mix 2, suggesting that the combination of herbal plants in Mix 2 may have attenuated the stress associated with the CD. It is also important to consider that factors such as age, breed, and environmental conditions may influence heterophile levels, making it difficult to predict the effects of fatty liver on heterophile counts in laying hens. Supporting this complexity, one study showed that feeding a diet containing lemon and garlic resulted in a reduced white blood cell count compared to a high-fat control diet [61].

A lower frequency of liver histopathology scores observed in hens fed the CD supplemented with Mix 2 compared to those without additives, suggests a potential protective role of the herbs in Mix 2 against FLHS development. Although the precise mechanisms underlying this protective effect remain unclear, the individual components of the herbal blend may contribute to FLHS prevention. In this regard, lemon and black pepper are recognized for their antioxidant and anti-inflammatory effects [12,13,78], which may help mitigate oxidative stress and inflammation—key factors in FLHS pathogenesis. Sumac, regulates lipid metabolism [11] and plays a role in preventing excessive fat accumulation in the liver. Additionally, dietary chicory has been associated with improved overall health and reduced liver histological changes [79], possibly due to its content of isoflavones, polyphenols, and antioxidants that may help reduce liver injury. However, existing data on the impact of medicinal plants on liver pathology are mixed. While some studies have found no significant effect of herbal essential oil mixtures on liver parameters [41,80], others have demonstrated the efficacy of specific herbal mixtures in reducing liver fat accumulation and inflammation in laying hens [52]. These discrepancies underscore the variability in biological responses to different herbal treatments. Trott et al. [4] noted varying degrees of hepatocellular vacuolization in FLHS-affected chickens, indicating that not all birds experience severe hepatic fat buildup. Similarly, Song et al. [33] reported that the severity of liver lesions may vary depending on factors such as the hens' age and dietary composition. These findings highlight the complexity of FLHS and the need for further research to better understand the individual and synergistic effects of these herbs on liver health and the progression of FLHS.

## Conclusion

The present experiment demonstrated that hens fed the CD exhibited reduced FI, EW, and EP rate, along with elevated blood levels of AST (indicating liver damage), and UA (indicating protein catabolism). Additionally, livers from the CD group showed higher histopathology scores consistent with FLHS. Mix 2 comprising sumac, lemon, black pepper, and chicory was the most effective among tested mixes. It improved performance parameters (FI, EW, and EP rate), lowered AST levels, and reduced liver histopathology scores in hens fed the CD. In contrast, the effects of other herbal mixtures were less consistent; some improved select parameters, while others showed no significant impact. Overall, the findings suggest that certain medicinal plant mixtures, particularly Mix 2, have the potential to alleviate the adverse effects of FLHS in laying hens. However, we have acknowledged key limitations of the study, including the relatively short duration, the

use of a single dose per herbal mixture, the lack of individual herb evaluation, and the restriction to one hen strain and age group, which may limit the broader applicability of the findings.

## Supporting information

**S1 File. Additional file.**
(DOCX)

**S2 File. Khodaei.**
(XLSX)

## Acknowledgments

The authors appreciate Razi University for supporting the current study.

## Author contributions

**Conceptualization:** Mohammadreza Khodaei, Mehran Torki, Fariborz Khajali.

**Data curation:** Mohammadreza Khodaei, Iraj Karimi.

**Formal analysis:** Mohammadreza Khodaei.

**Investigation:** Mohammadreza Khodaei, Mehran Torki, Iraj Karimi.

**Methodology:** Mohammadreza Khodaei, Mehran Torki, Iraj Karimi.

**Project administration:** Mehran Torki.

**Resources:** Mehran Torki.

**Software:** Mohammadreza Khodaei.

**Supervision:** Mehran Torki, Fariborz Khajali.

**Validation:** Mehran Torki, Fariborz Khajali.

**Writing – review & editing:** Mehran Torki, Fariborz Khajali, Iraj Karimi.

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
