## [Decision Letter · Decision Letter 0]

5 Jun 2025

Dear Dr. Torki,

We look forward to receiving your revised manuscript.

Kind regards,

Ewa Tomaszewska, DVM Ph.D

Academic Editor

PLOS ONE

Journal Requirements:

2. Please include captions for your Supporting Information files at the end of your manuscript, and update any in-text citations to match accordingly. Please see our Supporting Information guidelines for more information: http://journals.plos.org/plosone/s/supporting-information .

Reviewers' comments:

Reviewer's Responses to Questions

**Comments to the Author**

1. Is the manuscript technically sound, and do the data support the conclusions?

Reviewer #1: Yes

Reviewer #2: Partly

2. Has the statistical analysis been performed appropriately and rigorously?

Reviewer #1: Yes

Reviewer #2: Yes

3. Have the authors made all data underlying the findings in their manuscript fully available?

Reviewer #1: Yes

Reviewer #2: Yes

4. Is the manuscript presented in an intelligible fashion and written in standard English?

Reviewer #1: No

Reviewer #2: No

Reviewer #1: While the manuscript presents a well-designed experiment exploring the use of herbal additives to prevent FLHS in laying hens, the overall quality of English language and expression requires significant improvement. Numerous grammatical errors, awkward phrasings, and formatting inconsistencies—especially in the introduction and methods sections—may hinder reader comprehension. For instance, several sentences are either incomplete or improperly constructed, such as line 76–77, and citation formatting is occasionally unclear or broken. I recommend that the authors undergo thorough professional English language editing to enhance clarity, readability, and overall presentation before further consideration.

Reviewer #2: Dear Authors,

Strictly follow my comments/corrections in the attached file.

1. In the introduction section, need for the study should be highlighted at the end followed by the hypothesis of the study

2. In methodology, clearly elaborate the sample size determination

3. add appropriate citations

4. In the results, improve readability and add actual p-value of each result

5. In the discussion, add logical reasoning of each result before discussing with the previous studies.

6. In the conclusion, briefly describe and add limitation of the study

Regards

**Do you want your identity to be public for this peer review?** For information about this choice, including consent withdrawal, please see our Privacy Policy

Reviewer #1: No

Reviewer #2: **Yes: ** Sohail Ahmad

---

## [Author Response · Author response to Decision Letter 1]

1 Jul 2025

Reviewer #1: While the manuscript presents a well-designed experiment exploring the use of herbal additives to prevent FLHS in laying hens, the overall quality of English language and expression requires significant improvement. Numerous grammatical errors, awkward phrasings, and formatting inconsistencies—especially in the introduction and methods sections—may hinder reader comprehension. For instance, several sentences are either incomplete or improperly constructed, such as line 76–77, and citation formatting is occasionally unclear or broken. I recommend that the authors undergo thorough professional English language editing to enhance clarity, readability, and overall presentation before further consideration.

We have thoroughly reviewed the entire manuscript and carefully revised the text to address all grammatical errors, awkward phrasings, and formatting inconsistencies, particularly in the Introduction and Methods sections as noted. The sentence on lines 76–77 has been restructured for clarity. The citation formatting throughout the manuscript has been corrected. We have also sought assistance from a professional English language editing service to ensure the manuscript meets high standards of readability and presentation.

Line 37, the author could change “background” to “introduction”, because “background” may be confused as the same one in Abstract.

The heading “Background” was changed to “Introduction”.

Lines 38–42: The point that the liver synthesizes lipids and is heavily burdened in hens is repeated. The author could try to combine them with a single sentence.

We have revised the text to eliminate redundancy and merged the information into a concise sentence (page 2, lines 40-44).

The methods mentioned in the manuscript should be cited formally. Such as Line 51-54.

We have carefully reviewed the entire Methods section to ensure all procedures, including those mentioned in lines 51–54, are properly cited with relevant and appropriate references.

Lines 55–64: The paragraph listing all plants and their bio-actives is dense and hard to read. The author could Use a table or bullet points for readability or briefly summarize: "These herbs are rich in flavonoids, alkaloids, sulfur compounds, and dietary fibers, with demonstrated antioxidant, anti-inflammatory, and lipid-lowering properties."

Thank you for your helpful suggestion. We have replaced the dense paragraph with a table summarizing the medicinal plants used and their key bioactive components (Table 1).

Lines 37-69, the author could use clearer topic sentences to guide the reader through each section logically, including liver function, disease description, economic impact, and herbal solutions.

We have revised lines 37–69 to include clearer and more distinct topic sentences at the beginning of each major section.

Lines 88–90, the author use informal phrasing “running into the experimental trial", which can be corrected to "A two-week adaptation period was implemented to acclimate hens to the diets prior to the 12-week experimental trial."

We have revised the sentence to eliminate informal phrasing.

For animal study design, the randomization or blinding details were missing. Please mark those in the experiment design part.

Thank you for highlighting this important point. In response, we have updated the experimental design section to include details regarding randomization and blinding (page 5, lines 89-90).

No mention of sample size calculation or statistical power. The author could revise as "Sample size was determined based on expected effect sizes from previous studies to achieve 80% power at α = 0.05."

We have revised the Methods section to include a statement on sample size calculation. The text now reads: “Sample size was determined based on expected effect sizes from previous studies to achieve 80% power at α = 0.05” (page 5, lines 88-89).

Reviewer #2: Dear Authors, Strictly follow my comments/corrections in the attached file.

1. In the introduction section, need for the study should be highlighted at the end followed by the hypothesis of the study

Thank you for your insightful comment. We have revised the final part of the Introduction to clearly emphasize the need for the study and to better establish the context and significance of our research. Additionally, we have now explicitly stated the hypothesis of the study to guide the reader and clarify our research objectives.

2. In methodology, clearly elaborate the sample size determination

We have revised this section to provide a clear explanation of how the sample size was determined (Page 5, lines 88-89).

3. Add appropriate citations

We have thoroughly reviewed the entire Methods section and ensured that all procedures are now properly cited with relevant and appropriate references.

4. In the results, improve readability and add actual p-value of each result

Thank you for your valuable comment. We have thoroughly revised the Results section to enhance clarity and improve readability. Additionally, we have included the actual p-values for all statistical comparisons rather than simply stating significance levels (e.g., p < 0.05).

5. In the discussion, add logical reasoning of each result before discussing with the previous studies.

Thank you. We have revised the Discussion section to provide a clear and logical interpretation of each key result before comparing our findings with those of previous studies. These revisions aim to enhance the scientific depth and coherence of the discussion.

6. In the conclusion, briefly describe and add limitation of the study.

We have revised the Conclusion section to include a brief summary of the main findings, emphasizing the significance of our results. In addition, we have acknowledged the limitations of the study, such as the relatively short duration, the use of a single dose per herbal mixture, the lack of individual herb evaluation, and the restriction to one hen strain and age group. This addition provides a balanced perspective and suggests directions for future research.

Ad libitum feed was provided? at this age? why?

The CD was intentionally formulated to predispose hens to fatty liver hemorrhagic syndrome (FLHS). However, due to its imbalanced nutrient profile, feed intake could be influenced as hens attempt to compensate for nutrient deficiencies. By allowing ad libitum access, we enabled the hens to express natural compensatory feeding behavior, which is crucial for observing true diet-induced metabolic and hepatic changes.

---

## [Decision Letter · Decision Letter 1]

12 Jul 2025

Dear Dr. Torki,

Thank you for submitting your manuscript to PLOS ONE. After careful consideration, we feel that it has merit but does not fully meet PLOS ONE’s publication criteria as it currently stands. Therefore, we invite you to submit a revised version of the manuscript that addresses the points raised during the review process.

Please submit your revised manuscript by Aug 26 2025 11:59PM. If you will need more time than this to complete your revisions, please reply to this message or contact the journal office at plosone@plos.org . A rebuttal letter that responds to each point raised by the academic editor and reviewer(s). You should upload this letter as a separate file labeled 'Response to Reviewers'.A marked-up copy of your manuscript that highlights changes made to the original version. You should upload this as a separate file labeled 'Revised Manuscript with Track Changes'.An unmarked version of your revised paper without tracked changes. You should upload this as a separate file labeled 'Manuscript'.

We look forward to receiving your revised manuscript.

Kind regards,

Ewa Tomaszewska, DVM Ph.D

Academic Editor

PLOS ONE

Journal Requirements:

Reviewers' comments:

Reviewer's Responses to Questions

**Comments to the Author**

Reviewer #1: All comments have been addressed

Reviewer #2: All comments have been addressed

2. Is the manuscript technically sound, and do the data support the conclusions?

Reviewer #1: Yes

Reviewer #2: Partly

3. Has the statistical analysis been performed appropriately and rigorously?

Reviewer #1: Yes

Reviewer #2: Yes

4. Have the authors made all data underlying the findings in their manuscript fully available?

Reviewer #1: Yes

Reviewer #2: Yes

5. Is the manuscript presented in an intelligible fashion and written in standard English?

Reviewer #1: Yes

Reviewer #2: Yes

Reviewer #1: The manuscript is well written and addresses an important question in the field. The authors have responded thoroughly to all previous comments and have made the necessary revisions to improve the clarity, rigor, and impact of the work. I have no further concerns regarding scientific content, ethics, or publication practices. I recommend the manuscript for publication in its current form.

Reviewer #2: Dear Authors,

Follow my comments in the attached file, in the materials and methods section, give details about cage manufacturer and stocking density per bird. Also remove the model used for THI by Marai et al. as this study focuses on Rabbits and this model is applicable to different species. Remove this model and add the appropriate model suitable for chickens.

Improve the logical reasoning of the results. in conclusion, add limitations of the study

Regards

**Do you want your identity to be public for this peer review?** For information about this choice, including consent withdrawal, please see our Privacy Policy

Reviewer #1: No

Reviewer #2: **Yes: ** Sohail Ahmad

---

## [Author Response · Author response to Decision Letter 2]

25 Jul 2025

Reviewer #1: The manuscript is well written and addresses an important question in the field. The authors have responded thoroughly to all previous comments and have made the necessary revisions to improve the clarity, rigor, and impact of the work. I have no further concerns regarding scientific content, ethics, or publication practices. I recommend the manuscript for publication in its current form.

Thank you very much, indeed.

Reviewer #2: Dear Authors, Follow my comments in the attached file, in the materials and methods section, give details about cage manufacturer and stocking density per bird.

Thank you. The stocking density has been already provided in the Materials and Methods section, as stated in the following sentence:

“The hens were housed in wire cages (Machine Toyour, Co., Juybar, Mazandaran, Iran) measuring 45 × 45 × 45 cm, with three birds per cage, corresponding to a stocking density of 675 cm² per bird.”

The name of cage manufacturer has now been added (Page 5, Lines 91-92).

Also remove the model used for THI by Marai et al. as this study focuses on Rabbits and this model is applicable to different species. Remove this model and add the appropriate model suitable for chickens.

Thank you for bringing this up. We have replaced the previous THI model by a new THI model specifically used for poultry (Boonkum et al., 2025).

Improve the logical reasoning of the results.

Thank you for your comment. Based on the available literature, we have improved the logical reasoning of the results by clarifying the arguments, aligning our findings with previous studies, and strengthening the supporting evidence. Please see the changes in the revised manuscript.

In conclusion, add limitations of the study

Thank you for your comment. The limitation of the study had already been acknowledged in the conclusion section (Page 31, Lines 563-566).

---

## [Decision Letter · Decision Letter 2]

31 Jul 2025

Application of various mixtures of medicinal herbs in the diet of laying hens: Evaluating preventive approach of fatty liver syndrome

PONE-D-25-15950R2

Dear Dr. Mehran Torki,

We’re pleased to inform you that your manuscript has been judged scientifically suitable for publication and will be formally accepted for publication once it meets all outstanding technical requirements.

Kind regards,

Ewa Tomaszewska, DVM Ph.D

Academic Editor

PLOS ONE

Additional Editor Comments (optional):

Reviewers' comments:

Reviewer's Responses to Questions

**Comments to the Author**

Reviewer #2: All comments have been addressed

2. Is the manuscript technically sound, and do the data support the conclusions?

Reviewer #2: Yes

3. Has the statistical analysis been performed appropriately and rigorously?

Reviewer #2: Yes

4. Have the authors made all data underlying the findings in their manuscript fully available?

Reviewer #2: Yes

5. Is the manuscript presented in an intelligible fashion and written in standard English?

Reviewer #2: Yes

Reviewer #2: (No Response)

**Do you want your identity to be public for this peer review?** For information about this choice, including consent withdrawal, please see our Privacy Policy

Reviewer #2: No

---

## [Editor Report · Acceptance letter]

PONE-D-25-15950R2

PLOS ONE

Dear Dr. Torki,

I'm pleased to inform you that your manuscript has been deemed suitable for publication in PLOS ONE. Congratulations! Your manuscript is now being handed over to our production team.

Kind regards,

on behalf of

Professor Ewa Tomaszewska

Academic Editor

PLOS ONE